# Spatio-Temporal Distributions of the Land Use Efficiency Coupling Coordination Degree in Mining Cities of Western China

**Junfang Yuan [1], Zhengfu Bian [1,\*], Qingwu Yan [1] and Yuanqing Pan [2]**

[1]  School of Environment Science and Spatial Informatics, China University of Mining and Technology, Xuzhou 221116, China; JunfangYuan@cumt.edu.cn (J.Y.); yanqingwu@cumt.edu.cn (Q.Y.)

[2]  Land and Resources in Henan province Scientific Research Institute, Zhengzhou 450001, China; panyuanqing-110@163.com

\*  Correspondence: Zfbian@cumt.edu.cn; Tel.: +86-135-0521-5978

**Abstract:** A high coupling coordination degree of urban land use efficiency promotes sustainable regional economic development. In this study, land use efficiency coupling coordination degrees were calculated for 36 mining cities of western China, with a focus on economic, social, and ecological benefits for land use efficiency. Four years (2000, 2005, 2010, and 2015) of data were selected. A land use efficiency index system was generated and the improved entropy method was used to calculate the index weights of land use efficiency for each year. The spatial distributions of the coupling coordination degree were assessed by the ArcGIS spatial analysis tool. Spatial correlation analysis was conducted for the coupling coordination degree. The following conclusions could be drawn: (1) According to the composite index results, urban land use efficiency could be divided into three stages and showed several different time patterns in mining cities of western China; (2) analysis of the spatial and temporal distributions of the land use efficiency coupling coordination degree identified a low level of coupling coordination and reluctant coupling coordination. An obvious core-periphery and gradual trickle-down trend was observed; (3) the land use efficiency of western mining cities presents negative and positive spatial autocorrelation characteristics. Shizuishan city, Ordos city, Jinchang city, and Wuhai city have significant aggregation types. Therefore, the western mining cities were subjected to different complex time and space characteristics.

**Keywords:** land use efficiency; improved entropy method; coupling coordination degree; spatial and temporal distributions; mining cities; spatial autocorrelation

---

## 1. Introduction

The ultimate goal of land use is to achieve a comprehensive and coordinated development of economic, social, and ecological benefits of land use [1]. The realization of this goal is affected by regional natural resource endowment, the stage of social and economic development, the degree of land reclamation, and the quality of the ecological environment. Especially in the mining cities of western China, development focuses on and vigorously supports the development of regional urbanization and industrialization [2]. Clearly, an excessively fast pace of urbanization can lead to problems of land use in many respects, e.g., an accelerated decline of ecological quality, land degradation, an increasingly prominent contradiction between humans and land, and a drop of the overall urban land use benefitting level. This also hinders the regional qualitative growth.

Land resources can promote orderly urbanization, and thus, better promoting urban land use efficiency and achieving a high coupling coordination degree with balanced and healthy development

is an urgent issue of urbanization in China. Coupling of the coordination degree not only refers to the economic and social benefits of land utilization, but also includes ecological and environmental benefits. Land use efficiency is considered to be the sum of benefits of quaternity [3]. It refers to the sum of all benefits for cities obtained from land arrangement, utilization, and optimization in terms of both quantity and quality [4]. Moreover, it can optimize the spatial and temporal patterns of urbanization. An in-depth quantitative analysis of the land use efficiency coupling coordination degree [5] can provide a reference and analytical basis for both urbanization and land use policy making in China.

Current research on land use efficiency has shifted its focus from the common equilibrium theory and land use methods [6]. Currently, land use benefit evaluation [7], the effects of land use efficiency on the housing cost and income, the coupling coordination between industrialization and urbanization [8–10], the relationships between urban land use efficiency and the urbanization level [11–13], the land use extent, new urbanization, the land use structure, the land use efficiency, the spatial distribution, efficient and intensive urban land use [14], the coupling of urbanization economics and ecology [15–18], the coupling of industrial structure and land use efficiency [19], and the coupling of land use efficiency and economic development [20–25] are considered. Using urban planning and land rent theory, the concept of land use has been defined [26] and the importance of land use efficiency for resource user interests has been studied [27,28]. Within a particular study area (such as Bo hai Rim and Shaanxi Province [29]), the coupling relationship between urban land use efficiency and urbanization has been analyzed. The scope of research covers countries, provinces, cities, counties, urban areas, and urban agglomeration areas [30], and studies relating to sequence and time-node aspects. Recently, the coupling coordination between land use, ecological security patterns, and the environment has been studied [31–33]. Research methods mainly include the entropy method [34], the Analytic Hierarchy Process (AHP) [35,36], the coefficient of variation method [37], the Grey forecasting Model (GM(1,1)) [38], and principal component analysis (PCA) [39]. Urban land use efficiency analyses have been conducted [40]; however, ecological indicators have rarely been considered. However, the land use efficiency coupling coordination degree in the ecologically fragile western China remains unexplored.

This paper selected mining cities in western China, an area with a fragile environment and severe land degradation, where little research has been conducted on the land use efficiency coupling coordination degree. An improved entropy method and ecological indicators from a coupling coordination model were used to explore the land use efficiency coupling coordination degree in this region and its spatial distribution and spatial autocorrelation. The results highlight the current conditions and improve the planning and implementation of the economic development of western China's mining cities.

## 2. Data and Methods

### 2.1. Study Area

The longitude and latitude of eight provinces of western China are 73°40–126°04 E, 26°24–53°23 N. According to the Sustainable Development Planning of National Resource-Based Cities (2013-2020) issued by the State Council, there are 248 coal mining cities in China. Considering the availability and representativeness of the available data, autonomous prefectures and areas were removed and coal-mining areas were added. Finally, 36 prefecture-level mining cities from eight provinces (Inner Mongolia, Xinjiang, Yunnan, Guizhou, Sichuan, Shaanxi, Gansu, and Ningxia) (Figure 1) in western China were selected as research sites.

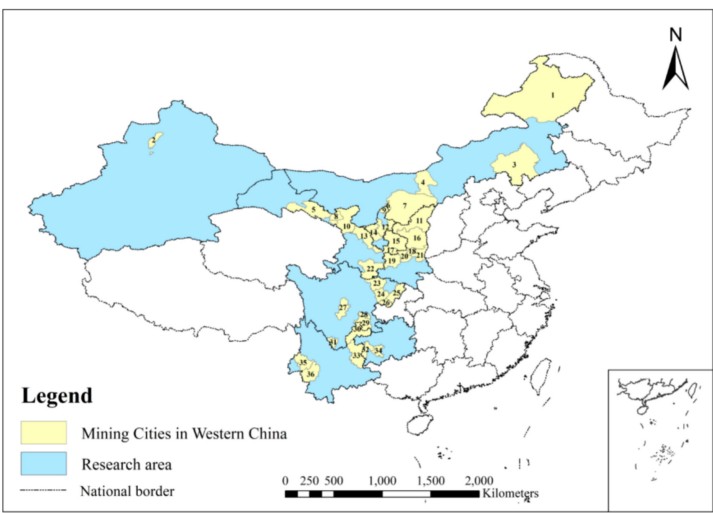

**Figure 1.** Location of the study areas. 1: Hulunbeir City; 2: Karamay City; 3: Chifeng City; 4: Baotou City; 5: Zhangye City; 6: Wuhai City; 7: Ordos City; 8: Jinchang City; 9: Shizuishan City; 10: Wuwei City; 11: Yulin City; 12: Wuzhong City; 13: Silver City; 14: Zhongwei City; 15: Qingyang City; 16: Yan'an City; 17: Pingliang City; 18: Tongchuan City; 19: Baoji City; 20: Xianyang City; 21: Weinan City; 22: Longnan City; 23: Guangyuan City; 24: Nanchong City; 25: Dazhou City; 26: Guang'an City; 27: Ya'an City; 28: Zigong City; 29: Yi bin City; 30: Zhaotong City; 31: PanzhiHua City; 32: Liu panshui City; 33: Qujing City; 34: An shun City; 35: Baoshan City; 36: Lincang City.

## 2.2. Data Source

In the ecological benefit system, the utilization of industrial solid waste, innocuous treatment rate of domestic waste, centralized processing rate of wastewater treatment plants, amount of industrial wastewater discharge, and amount of industrial sulfur dioxide emissions originate from the statistical bulletins for the national economic and social development of each prefecture-level city for the relevant years. The green coverage rate of built-up area and park green area originate from China City Statistical Yearbooks.

With regard to social and economic benefits, all data were obtained from the China City Statistical Yearbooks for 2001, 2006, 2011, and 2016, for eight western provinces (Shaanxi, Gansu, Ningxia, Yunnan, Guizhou, Sichuan, Inner Mongolia, and Xinjiang).

## 2.3. Construction of the Evaluation Index System

Specific to the index level, the selection of indexes should be reasonable, be able to explain the coupling relationship, have the characteristics of representativeness, be easy to handle, and be easy to obtain. For this study, all land areas under the jurisdiction of typical mining cities in western China were selected as the boundary of research areas. Various evaluation dimensions were treated in a quantitative manner, to more accurately measure the comprehensive benefits and coupling coordination degree of land use in cities. The existing research results were mainly obtained from economic benefits and social benefits, while this paper also considers ecological and environmental benefits, mainly as the park green area, green coverage rate, standard-reaching rate of industrial solid waste, domestic waste, wastewater treatment plant and industrial wastewater discharge, and amount of emissions of industrial sulfur dioxide. These factors comprehensively reflect the ecological utilization of urban land and the environmental benefits of urban land use. With regard to social benefits, indexes were mainly selected based on urban public services and infrastructure related to people's social life, including the number of collections in public libraries, the number of beds, and the number of public buses [41].

With industrialization and the gradual evolution of tertiary industry into the follow-up power of technological progress and economic development (especially the tertiary industry's gradually increased driving force for economic development), the proportion of the tertiary industry for GDP

and the added value of the secondary industry and the tertiary industry per km were selected as a measure of economic benefits. This is in line with historical law and the development trend of economic development and urbanization.

Index selection also considered the developmental stages and resource endowments of mining cities in western China. These indexes reflect the internal operation mechanisms of land use efficiency. Prior studies only focused on land use inputs and outputs, which cannot expose land use efficiency internal coupling coordination mechanisms [42]. Therefore, social, economic, and ecological factors should also be integrated into the evaluation of land use efficiency in mining cities of western China (Table 1).

**Table 1.** Index system for evaluating the land use efficiency in mining cities of western China.

| Criteria Layer | NO. | Index Layer | Attribute | Weight 2000 | Weight 2005 | Weight 2010 | Weight 2015 |
|---|---|---|---|---|---|---|---|
| Economic benefits | X1 | GDP per square kilometer (ten thousand Yuan/km$^2$) | + | 0.0635 | 0.0749 | 0.0929 | 0.0889 |
| | X2 | Proportion of the tertiary industry for GDP (%) | + | 0.0113 | 0.0189 | 0.0142 | 0.0409 |
| | X3 | Total fixed assets investment per square kilometer (ten thousand Yuan/km$^2$) | + | 0.0402 | 0.0867 | 0.0766 | 0.0782 |
| | X4 | Added value of the secondary and tertiary industries per square kilometer (a hundred million Yuan/km$^2$) | + | 0.1231 | 0.08 | 0.0854 | 0.0937 |
| | X5 | Total profit of industrial enterprises per square kilometer (ten thousand Yuan/km$^2$) | + | 0.1044 | 0.1737 | 0.1475 | 0.0521 |
| Ecological benefits | X6 | Utilization of industrial solid waste (%) | + | 0.032 | 0.0332 | 0.0216 | 0.021 |
| | X7 | Green coverage rate of built-up area (%) | + | 0.023 | 0.026 | 0.0148 | 0.0142 |
| | X8 | Innocuous treatment rate of domestic waste (%) | + | 0.0213 | 0.0265 | 0.0245 | 0.0138 |
| | X9 | Park green area (hectare) | + | 0.0714 | 0.0846 | 0.0695 | 0.0613 |
| | X10 | Centralized processing rate of wastewater treatment plant (%) | + | 0.034 | 0.0382 | 0.0287 | 0.0085 |
| | X11 | Amount of industrial wastewater discharge (ten thousand tons/km$^2$) | − | 0.0146 | 0.0151 | 0.0139 | 0.0067 |
| | X12 | Amount of industrial sulfur dioxide emission (tons/km$^2$) | − | 0.0055 | 0.0058 | 0.0067 | 0.0062 |
| Social benefits | X13 | Population density (people/km$^2$) | − | 0.012 | 0.0159 | 0.0175 | 0.0182 |
| | X14 | Urban road area per capita (m$^2$) | + | 0.0481 | 0.0444 | 0.0814 | 0.1001 |
| | X15 | Total number of public library collections per ten thousand people (one thousand volumes or copies/ten thousand people) | + | 0.1922 | 0.0852 | 0.0936 | 0.1859 |
| | X16 | The number of college students per ten thousand people (students/ten thousand people) | + | 0.1011 | 0.0928 | 0.1142 | 0.1027 |
| | X17 | The number of hospital beds per ten thousand people (beds/ten thousand people) | + | 0.0184 | 0.0432 | 0.0502 | 0.0464 |
| | X18 | The number of public buses per ten thousand people (buses/ten thousand people) | + | 0.0839 | 0.0551 | 0.0469 | 0.0612 |

Note: "−" represents a negative effect on land use efficiency; "+" represents a positive effect on land use efficiency.

### 2.4. Data Standardization and Weight Determination

The evaluation indexes for different fields differed in terms of their attributes. The differences between index values were large. To minimize the possible interference caused by these factors, the standard deviation method was used to standardize original data [43].

$$X_{ij}' = \left(x_{ij} - \overline{x_j}\right)/s_j. \tag{1}$$

In Formula (1), $X_{ij}$ represents the original value of index j of city i, $X_j$ represents the mean value of index j, and $S_j$ represents the standard deviation of index j.

With MATLAB software, the improved entropy method can be adopted to improve the evaluation indexes. By transforming the data after standardization, the influence of the negative number can rationally be eliminated. After calculation, the range is from −4 to 4. The coordinates can be translated as follows:

$$Z_{ij} = X_{ij}' + 4 \tag{2}$$

On the basis of Formula (2), the data after translation are normalized:

$$f_{ij} = Z_{ij}/\sum_{i=1}^{n} Z_{ij}.(i = 1, 2, \ldots, n; j = 1, 2, \ldots m) \tag{3}$$

where $f_{ij}$ represents the value after index standardization.

The information entropy $H_j$ of index j is calculated as follows:

$$H_j = -K\sum_{i=1}^{n}\left(f_{ij} \times \ln f_{ij}\right), K = (\ln n)^{-1}, (i = 1, 2, \ldots, n; j = 1, 2, \ldots m) \tag{4}$$

The information weight $w_j$ of index j is calculated as follows:

$$w_j = \frac{1 - H_j}{m - \sum\limits_{j=1}^{m} H_j} \tag{5}$$

### 2.5. Evaluation Index Calculation

After the standardization processing and weight assignment of original data of the indexes, the linear weighted method was used to evaluate the overall level of the three land use efficiency sub-systems.

$$U_t = \sum_{j=1}^{m} f_{ij}w_j(t = 1, 2, 3) \tag{6}$$

In Formula (6), U1, U2, and U3 represent the levels of the three land use efficiency sub-systems (society, economy, and environment, respectively), $f_{ij}$ represents the value after standardization processing of index $X_{ij}$ in the evaluation system (criteria layer and index layer), $w_j$ represents the index weight, and m represents the number of indexes in the evaluation system.

### 2.6. Coupling Coordination Model

The term coupling is derived from physics, and indicates that two or more systems or motion forms interact with each other through key factors, such as coupling elements and operation mechanisms, finally leading to coordination. Coordination refers to the cooperation and virtuous cycle between systems or elements [44].

Based on existing research results, a coupling coordination model of land use efficiency was constructed. Coupling is used to describe the interactions and synergies between systems. The coupling

coordination degree indicates the strengths and weaknesses of these interactions [45]. It is a measure used to promote both the synergy and coordinated action of the system to an orderly mechanism. The interaction between and influence of elements of the three sub-systems (society, economy, and environment) of land use efficiency in this study are defined as the coupling of the land use efficiency of mining cities of western China. The coupling coordination model for land use efficiency and mechanisms is presented in Figure 2.

In this study, "coupling coordination" refers to the interaction between the elements of three sub-systems (economy, society, and ecology) [46,47] in the land use efficiency of mining cities of western China under complex mechanisms (Figure 2).

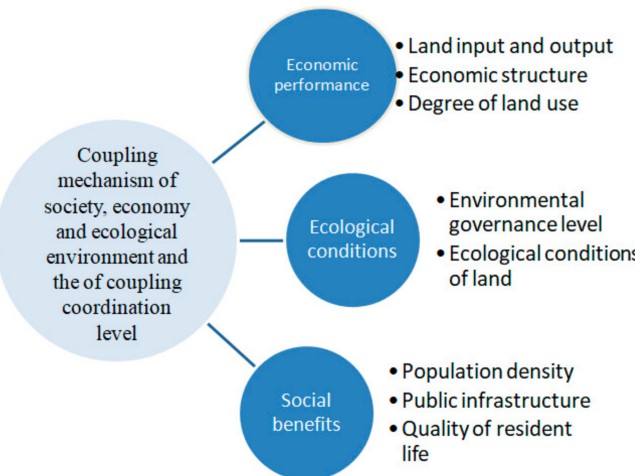

**Figure 2.** Land use efficiency coupling coordination mechanisms.

A coupling coordination model measures the degree of integration between land use efficiency and the urbanization level of mining cities in western China.

$$C = \left[ \frac{U1 * U2 * U3}{[U1 + U2 + U3]^3} \right]^{1/3} \tag{7}$$

$$T = \alpha U_1 + \beta U_2 + \gamma U_3 \tag{8}$$

$$D = \sqrt{C * T} \tag{9}$$

In Formula (7), $0 \le C \le 1$. If $C = 1$, adaptability occurs among the three, which develop in an orderly direction, n; if $C = 0$, the three are in a disordered state. In Formulas (8) and (9), T represents the land use efficiency. $\alpha$, $\beta$, and $\gamma$ are undetermined coefficients, which satisfy $\alpha + \beta + \gamma = 1$. The entropy method was applied for objective and accurate evaluation. MATLAB software was used to calculate the following evaluation indices: $\alpha$, $\beta$, and $\gamma$, which were 0.4557, 0.3425, and 0.2018, respectively, in 2000; 0.3366, 0.4342, and 0.2292, respectively, in 2005; 0.4038, 0.4166, and 0.1796, respectively, in 2010; and 0.5145, 0.3538, and 0.1317, respectively, in 2015. D represents the coupling coordination degree of economy, society, and ecology. C and T ranged from (0, 1], $D \in (0, 1]$. The higher the value of D, the higher the coordination degree.

The coupling degree was divided into four stages. According to relevant literature and mining city development, a hierarchy was defined using the natural breakpoint method [48], so that the coupling levels were graded as shown in Table 2.

**Table 2.** Division of the coupling coordination stages of land use efficiency.

| Coupling Coordination Degree (D) | 0<*D*≤0.3 | 0.3<*D*≤0.5 | 0.5<*D*≤0.8 | 0.8<*D*≤1 |
|---|---|---|---|---|
| Coupling coordination stage | Reluctant coupling coordination | A low level of coupling coordination | Moderate coupling coordination | A high level of coupling coordination |

*2.7. Spatial Autocorrelation Analysis*

Spatial autocorrelation analysis was conducted to assess the correlation between the common geographical features or attribute values in a regional unit and its neighbors [49], which can be analyzed from both global and local perspectives. Global Moran's I was selected to measure the spatial level of land use efficiency coupling coordination. The local LISA index was adopted to describe the local spatial heterogeneity characteristics of land use efficiency coupling coordination, which is conducive to a scientific and reasonable analysis of its internal spatial correlation rules on different spatial units [50].

$$I = \frac{\sum\limits_{i=1}^{n}\sum\limits_{j=1}^{n} W_{ij}(D_i - \overline{D})(D_j - \overline{D})}{S^2 \sum\limits_{i=1}^{n}\sum\limits_{j=1}^{n} W_{ij}} \tag{10}$$

$$S^2 = \frac{1}{n}\sum\limits_{i=1}^{n}(D_i - \overline{D})^2 \tag{11}$$

$$I_i = \frac{(D_i - \overline{D})}{S^2}\sum\limits_{j=1}^{n} W_{ij}(D_j - \overline{D}) \tag{12}$$

In Formulas (10) and (12), $D_i$ and $D_j$ represent the coupling coordination degree of space units i and j, respectively; $\overline{D}$ represents the mean value of the coupling coordination degree of space units i and j, respectively; n represents the number of units in the observed space; $W_{ij}$ represents the spatial weight coefficient matrix; $S^2$ represents the variance of the coupling coordination degree of each spatial element; I represents the global Moran's I; and $I_i$ represents the partial LISA index.

**3. Results and Discussion**

*3.1. Measure of Comprehensive Level of Land Use Efficiency*

Based on the index data processing method and the land use efficiency measure model, the comprehensive evaluation index values for the land use efficiency of mining cities of western China were calculated for 2000, 2005, 2010, and 2015 (Figure 3). Furthermore, the evolution of the comprehensive level of land use efficiency was explored.

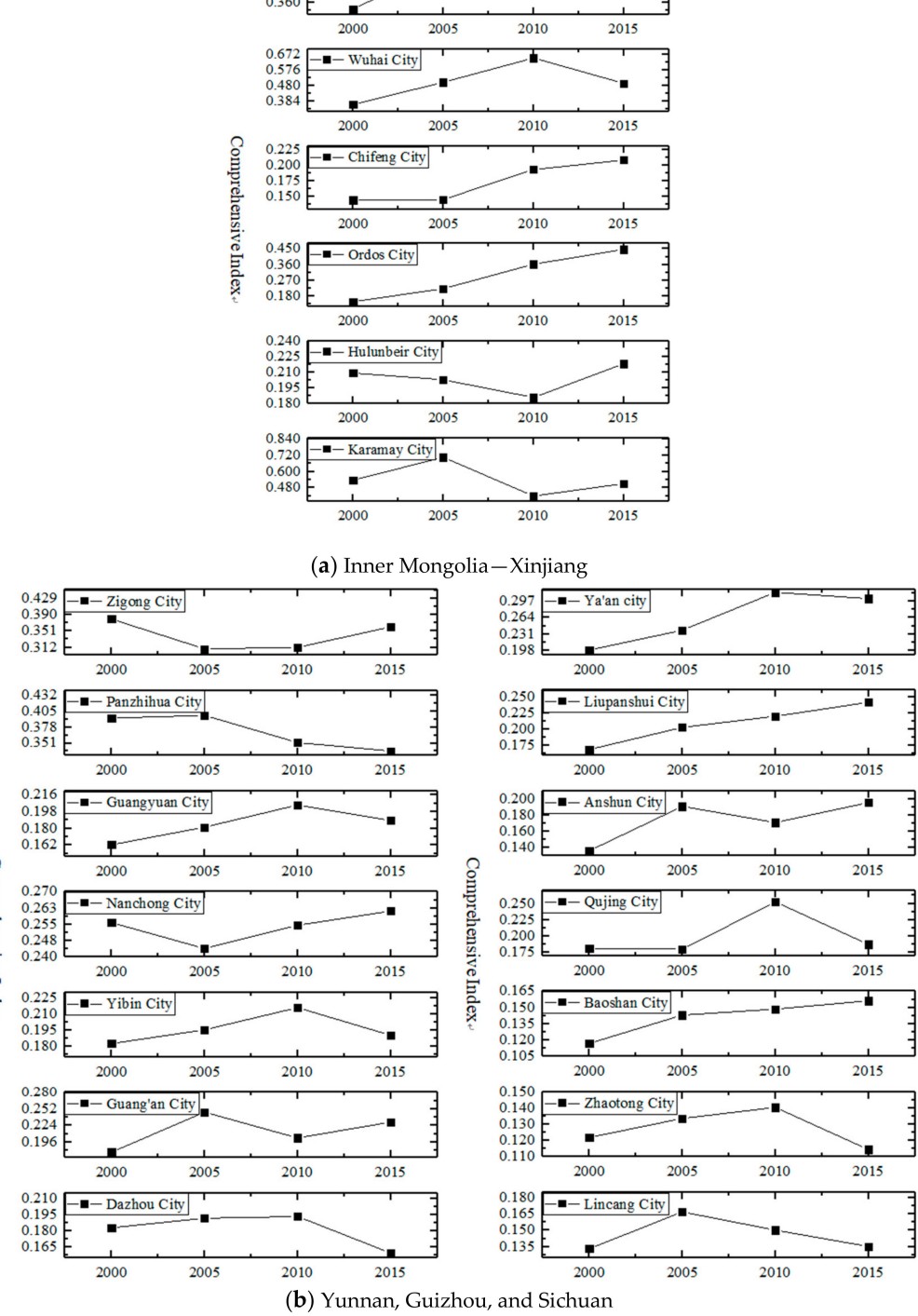

(**a**) Inner Mongolia—Xinjiang

(**b**) Yunnan, Guizhou, and Sichuan

**Figure 3.** *Cont.*

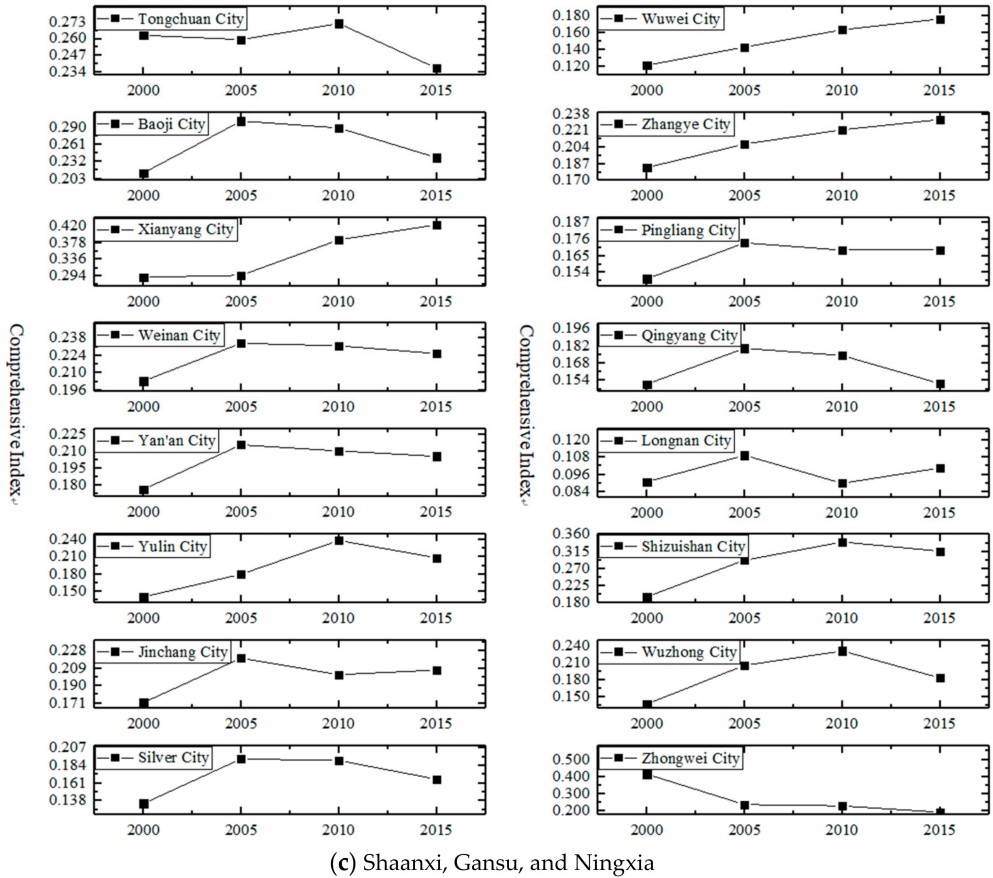

(**c**) Shaanxi, Gansu, and Ningxia

**Figure 3.** Accumulation map of land use efficiency over time in mining cities of western China.

First, the overall land use efficiency showed several different time patterns of land use efficiency in mining cities of western China. With regard to the comprehensive level of land use efficiency, Xianyang city of Shaanxi and Erdos city of Inner Mongolia showed a gradual increasing trend; Ya'an city of Sichuan, Xianyang city of Shaanxi, and Liupanshui city of Guizhou showed a first decreasing and then increasing trend; Wuhai city and Baotou city of Inner Mongolia, Panzhihua city of Sichuan, and Zhongwei city of Shaanxi showed a first increasing and then decreasing trend; and Karamay city showed a first increasing, then decreasing, and finally increasing trend.

Secondly, the comprehensive level of land use efficiency in mining cities of western China was divided into three stages. Stage one included Erdos city, Wuhai city, Karamay city, Panzhihua, and Xianyang, with efficiencies above 0.4. Stage two included Zigong, Baotou, Ya'an, and Tong chuan city, with a medium level of land use efficiency that was below that of stage one and with low variability, ranging from 0.2 to 0.4. Stage three included Lincang city, Zhao tong city, Baoshan city, and Longnan city, with a lower land use efficiency and variability, ranging from 0.1 to 0.2. Among these cities, there was a small gap in the comprehensive level of land use efficiency.

In previous studies, the traditional entropy method has often led to a deviation of the evaluation results due to extreme or negative values in the evaluation [51]. In this study, compared with traditional methods, an improved entropy method was applied to measure the land use benefit coupling coordination degree in mining cities of western China, which compensated for the defect of the traditional entropy method through standardization processing. This method can objectively and effectively analyze the information relevance between index variables and avoid the limitation of subjective weighting to a great extent. In addition, the traditional method is usually represented by a radar chart. In this study, Origin software was used to draw the accumulation map of land use efficiency. This meant that it was easy to present an overview of the evolutionary character, and analyze the relationship between social benefits, economic benefits, and environmental benefits.

### 3.2. Measured and Spatio-Temporal Distribution of the Land Use Efficiency Coupling Coordination Degree

#### 3.2.1. Coupling Coordination Degree

Based on the results of Figure 3, the coupling coordination degree results for mining cities of western China are shown in Table 3, as a result of combining them with the evaluation model of coupling coordination.

**Table 3.** Coupling coordination degree of mining cities of western China.

| City | 2000 | 2005 | 2010 | 2015 |
| --- | --- | --- | --- | --- |
| | Coupling Coordination Degree | Coupling Coordination Degree | Coupling Coordination Degree | Coupling Coordination Degree |
| Baotou City | 0.2986 | 0.4078 | 0.5503 | 0.3754 |
| Wuhai City | 0.3387 | 0.4026 | 0.4945 | 0.392 |
| Chifeng City | 0.1866 | 0.1848 | 0.3534 | 0.2181 |
| Ordos City | 0.2143 | 0.2466 | 0.4465 | 0.3745 |
| Hulunbeir City | 0.2297 | 0.1978 | 0.3032 | 0.2389 |
| Zigong City | 0.3258 | 0.3167 | 0.4013 | 0.3154 |
| PanzhiHua City | 0.3684 | 0.3575 | 0.4378 | 0.3326 |
| Guangyuan City | 0.1961 | 0.2108 | 0.3783 | 0.2193 |
| Nanchong City | 0.2718 | 0.2657 | 0.4081 | 0.2581 |
| Yi bin City | 0.2288 | 0.2405 | 0.386 | 0.2201 |
| Guang'an City | 0.1926 | 0.2441 | 0.3238 | 0.2037 |
| Dazhou City | 0.2145 | 0.2341 | 0.364 | 0.1808 |
| Ya'an City | 0.2486 | 0.2748 | 0.4227 | 0.3074 |
| Liu panshuiCity | 0.2069 | 0.2438 | 0.3802 | 0.2675 |
| An shun City | 0.1953 | 0.2155 | 0.3482 | 0.2235 |
| Qujing City | 0.2074 | 0.2307 | 0.4214 | 0.2179 |
| Baoshan City | 0.1744 | 0.1869 | 0.3178 | 0.204 |
| Zhaotong City | 0.1873 | 0.1752 | 0.304 | 0.162 |
| Lincang City | 0.1842 | 0.1834 | 0.3058 | 0.1809 |
| Tongchuan City | 0.2788 | 0.2742 | 0.4319 | 0.2603 |
| Baoji City | 0.2498 | 0.3013 | 0.4481 | 0.2619 |
| Xian yang City | 0.2836 | 0.3136 | 0.476 | 0.3608 |
| Weinan City | 0.2458 | 0.2533 | 0.3958 | 0.2393 |
| Yan'an City | 0.2347 | 0.2519 | 0.3709 | 0.2325 |
| Yulin City | 0.2144 | 0.2277 | 0.405 | 0.2373 |
| Jinchang City | 0.2423 | 0.2578 | 0.3476 | 0.2404 |
| Silver City | 0.2033 | 0.2193 | 0.3484 | 0.2089 |
| Wuwei City | 0.1874 | 0.1796 | 0.309 | 0.2088 |
| Zhangye City | 0.2281 | 0.2009 | 0.3358 | 0.254 |
| Pingliang City | 0.2161 | 0.217 | 0.3408 | 0.2131 |
| Qingyang City | 0.222 | 0.2224 | 0.3327 | 0.2074 |
| Longnan City | 0.1706 | 0.1646 | 0.2516 | 0.168 |
| Shizuishan City | 0.2254 | 0.3001 | 0.4742 | 0.3093 |
| Wuzhong City | 0.1997 | 0.2181 | 0.3663 | 0.2219 |
| Zhongwei City | 0.3075 | 0.2307 | 0.3688 | 0.2213 |
| Karamay City | 0.4331 | 0.4837 | 0.513 | 0.3849 |

According to the grading criteria listed in Table 1, the coupling coordination degree of the mining cities of western China is in the phase of reluctant coupling coordination, and shows a low level of coupling coordination and moderate coupling coordination. According to the sequence of the coupling coordination degree, the coupling coordination degree was divided into three types (Table 3): ① Reluctant coupling coordination. In 2000, 2005, 2010, and 2015, there were 31, 28, 1, and 28 cities in this group, accounting for 86.11%, 77.78%, 2.8%, and 77.78%, respectively; ② low coupling coordination. In 2000, 2005, 2010, and 2015, there were 5, 8, 33, and 8 cities, accounting for 13.89%, 22.22%, 91.67%,

and 22.22%, respectively; ③ moderate coupling coordination. In 2000, 2005, 2010, and 2015, there were 0, 0, 2, and 0 cities, respectively, without any city with a high level of coupling coordination. It can be seen that the land use efficiency coupling coordination degree of mining cities of western China is mainly in the stage of a reluctant and low level.

### 3.2.2. Spatial Distribution of Mining Cities

The results of Table 3 and Figure 4 show the coupling coordination degree spatial distributions, which were combined with the ArcGIS spatial analysis tool.

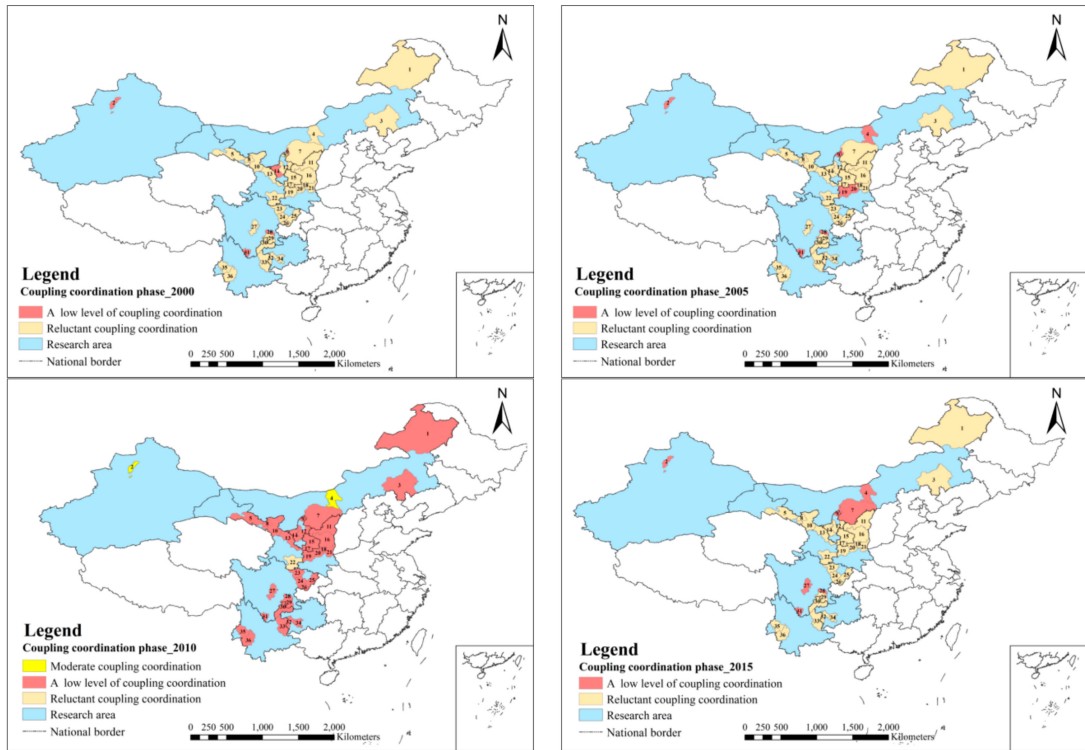

**Figure 4.** Coupling coordination degree spatio-temporal distributions for western China's mining cities in 2000, 2005, 2010, and 2015. In 2000 (red areas), 1: Hulunbeir City, 6: Wuhai City, 14: Zhongwei City, 28: Zigong City, and 31: PanzhiHua City; in 2005 (red areas), 2: Karamay City, 4: Baotou City, 6: Wuhai City, 7: Ordos City, 9: Shizuishan City, 28: Zigong City, and 31: PanzhiHua City; in 2010 (bright yellow areas), 2: Karamay City and 4: Baotou City; (yellow areas) 22: Longnan City; in 2015 (red areas), 2: Karamay City, 4: Baotou City, 6: Wuhai City, 7: Ordos City, 9: Shizuishan City, 27:Ya'an City, and 31: PanzhiHua City.

The spatial distributions are shown in Figure 4. The spatial differentiation features of the land use efficiency coupling coordination degree of mining cities of western China are as follows:

(1)　Reluctant coupling coordination: From 2000 to 2005 and from 2010 to 2015, Zhongwei city decreased from a low level of coupling coordination to reluctant coupling coordination; from 2010 to 2015, Baoji city decreased from low coupling coordination to reluctant coupling coordination; in 2000, Baotou city was at a reluctant coupling coordination level; Chifeng city was at a reluctant coupling coordination level in 2000 and 2005; Chifeng city was at a reluctant coupling coordination level in 2015; Erdos city was at a reluctant coupling coordination level from 2000 to 2005; and in 2015, Zhaotong was at a reluctant coupling coordination level;

(2)　Low level of coupling coordination: During the research period, the number of low-coupling coordination cities in the research units of mining cities of western China experienced a process of first increasing and then decreasing. From 2000 to 2015, Shizuishan city increased from a reluctant

coupling coordination level to a low level of coupling coordination; from 2010 to 2015, Zhongwei city increased from a reluctant coupling coordination level to a low level of coupling coordination; from 2000 to 2015, Baoji city increased from a reluctant coupling coordination level to a low level of coupling coordination; from 2010 to 2015, Xian Yang city increased from a reluctant coupling coordination level to a low level of coupling coordination; from 2000 to 2005, Panzhihua city and Zigong city were at a low coupling coordination level; from 2015, Ya'an city, Panzhihua city, and Zigong city were at a low coupling coordination level; from 2000 to 2015, Wuhai city and Hulunbeier city were at a low coupling coordination level; in 2005, Baotou city was at low-level coupling coordination; in 2015, Baotou city was at a moderate coupling coordination level and Chifeng city was at a low-level coupling coordination level; and Erdos city was at a low coupling coordination level from 2010 to 2015;

(3) Moderate coupling coordination: Baotou city and Karamay city achieved moderate coupling coordination levels in 2010. The discussion is based on the above analysis of the coupling coordination degrees, which were mainly dominated by reluctant and low levels. The coupling coordination degree presented three types (reluctant, low, and moderate), and no city was in the phase of high-level coupling coordination. The radiation driving effect of the core city is obvious.

The previous studies have mainly presented the object of the spatio-temporal distribution [52]. In this paper, with the ArcGIS spatial analysis software and coupling coordination model, the spatial-temporal differentiation characteristics of the coupling coordination level of land use benefits in western mining cities have been explored. The paper also studies the degree of economic, social, and ecological coupling coordination of the factors influencing the coupling coordination level of land use benefit in each research unit. The importance of the ecological and environmental benefits of land use, and the significance of carrying out the comprehensive, sustainable, and harmonious development land use in terms of economy, society, and ecology, have also been demonstrated.

### *3.3. Spatio-Temporal Correlation Analysis of the Land Use Efficiency Coupling Coordination Degree*

3.3.1. Global Spatial Autocorrelation Analysis of the Land Use Efficiency Coupling Coordination Degree

GeoDA analysis software was used to conduct a spatial exploration of land use benefits in mining cities of western China. Global spatial autocorrelation of land use benefits was conducted by employing Global Moran's I, which is helpful for assessing the spatial correlation characteristics of land use efficiency coupling coordination (Figure 5). The transverse axis of the scatter plot corresponds to the description variables, and the vertical axis corresponds to the space lag vector. The Moran scatter diagram consists of four quadrants, each with different attributes of spatial autocorrelation. The I quadrant (H - H) and the III quadrant (L - L) represent a positive correlation, and present similar eigenvalues of space, indicating that the eigenvalues of the units are high (low). However, the adjacent area unit eigenvalues are high (low). Usually, this is called the first I and III quadrant hot and cold spots. The II quadrant (L - H) and the IV quadrant (H - L) are negatively related, so the space unit characteristic value is low (high). However, the adjacent space unit characteristic values are high (low) [53]; if the eigenvalues are evenly distributed in the four quadrants, this indicates that no spatial autocorrelation exists. Objects without spatial association have been deleted, such as Karamay city in Xinjiang. The spatial distribution of the scatter diagram is as shown in the figure below.

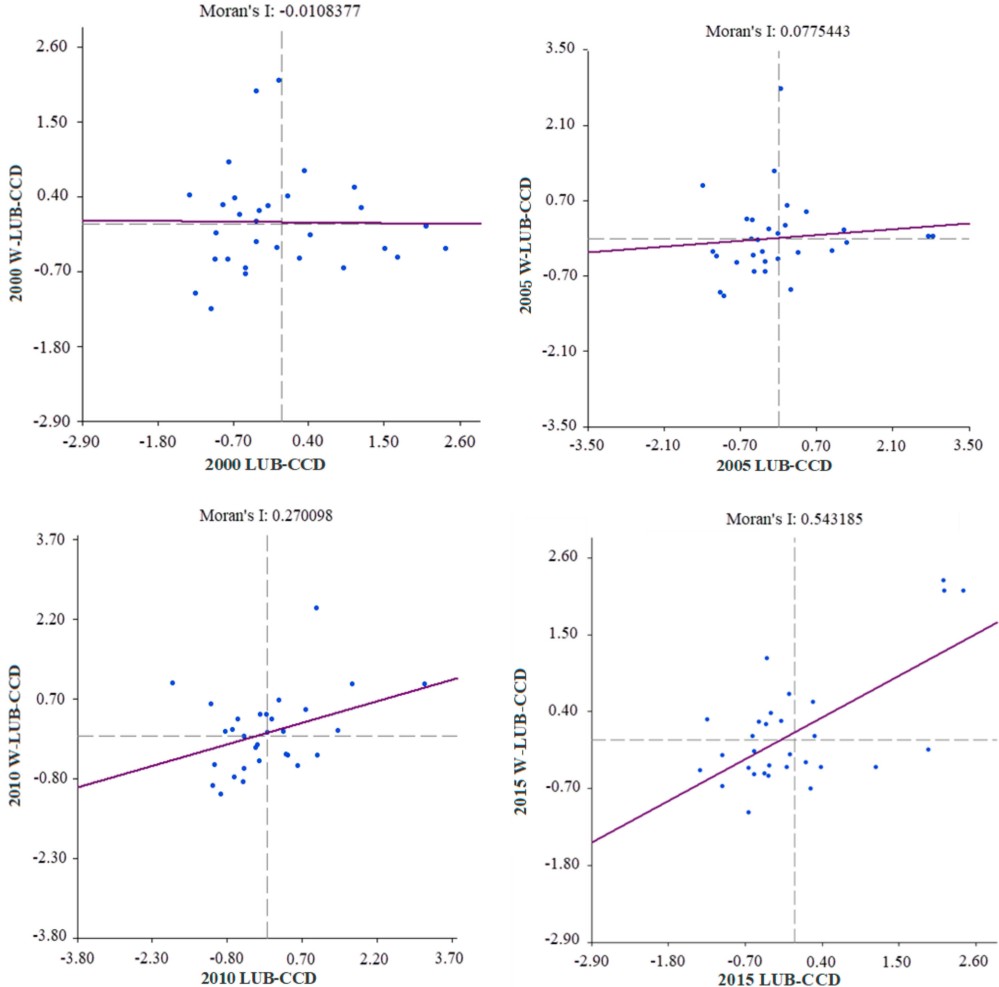

**Figure 5.** The land use benefit of the coupling coordinated scatter plot in mining cities of western China. (Horizontal axis) LUB-CCD represents the land use benefit of the coupling coordinated degree; (vertical axis) W- LUB-CCD is the spatial lag factor for the land use benefit of the coupling coordinated degree.

Four nodes in Nanchong of the first IV quadrant have a higher coupling coordination level and maintained the "high-low" aggregation type, and the coupling coordination level of its adjacent cities is generally low; four nodes are all in Guangyuan city, Pingliang city, and Longnan city. The first II quadrant and its coupling coordination level is low, basically showing a "low-high" aggregation type. However, its adjacent coupling coordination level is relatively high. In addition to Tongchuan city, Baotou city, Erdos city, and Wuhai city are in the first I quadrant (hot spots) (Table 4), indicating that the coupling coordination degree level was relatively high for 2005, 2010, and 2015. Baoshan city, Lincang city, Qujing city, Zhaotong city, and Wuwei city are all in the first III quadrant (cold spot area). Wuzhong city, Zhongwei city, and Silver city were also in the cold spots area in 2005, 2010, and 2015. This indicates that the coupling coordination level is low and the adjacent city's coupling coordination level is also low.

The cities of Moran's I values at four nodes from a negative correlation to positive correlation indicate a gradually increasing level of economic development in mining cities of western China. This aggregation characteristic changes gradually from cities with a low coupling coordination level to those with high coupling coordination. The cities with a high economic development level play a radiating and driving role. Each research unit adjacency relationship is gradually optimized from city to city. However, the number of cities gradually increased in the first III quadrant (cold spot area).

This shows that the radiation effect depends on a few cities, such as Baotou city, Ordos city, Wuhai city, and Tongchuan city, which play a radiating and driving role (Table 5).

**Table 4.** Distribution of first I and II quadrants in mining cities of western China.

| The I Quadrant | | | | The II Quadrant | | | |
|---|---|---|---|---|---|---|---|
| **2000** | **2005** | **2010** | **2015** | **2000** | **2005** | **2010** | **2015** |
| Tongchuan | Baotou | Baotou | Baotou | Wuzhong | Dazhou | Zhaotong | An shun |
| Xianyang | Ordos | Ordos | Ordos | Ordos | Guangyuan | Liu panshui | Guangyuan |
| Weinan | Wuhai | Wuhai | Wuhai | Guangyuan | Guang'an | An shun | Yi bin |
| Yan'an | Tongchuan | Yi bin | Tongchuan | Guang'an | Yi bin | Guangyuan | Pingliang |
| Longnan | Xian yang | Zigong | Baoji | Silver | Pingliang | Guang'an | Qingyang |
| | Weinan | Tongchuan | | Yi bin | Longnan | Pingliang | Longnan |
| | Yan'an | Xian yang | | Qingyang | Qingyang | Qingyang | Weinan |
| | | Weinan | | Pingliang | | Longnan | Yan'an |
| | | | | Longnan | | Yan'an | |
| □ | □ | □ | □ | □ | □ | □ | □ |

**Table 5.** Distribution of the first III and IV quadrants of mining cities of western China.

| The III Quadrant | | | | The IV Quadrant | | | |
|---|---|---|---|---|---|---|---|
| **2000** | **2005** | **2010** | **2015** | **2000** | **2005** | **2010** | **2015** |
| Baoshan | Wuzhong | Wuzhong | Wuzhong | Zhongwei | Nanchong | Qujing | Liupanshui |
| Lincang | Zhongwei | Zhongwei | Zhongwei | Baotou | Zigong | Nanchong | Nanchong |
| Qujing | Baoshan | Baoshan | Baoshan | Wuhai | Jinchang | Baoji | Zigong |
| Zhaotong | Lincang | Lincang | Lincang | Nanchong | Baoji | Yulin | Zhangye |
| Liupanshui | Qujing | Dazhou | Qujing | Zigong | | | Xian yang |
| An shun | Zhaotong | Jinchang | Zhaotong | Jinchang | | | |
| Wuwei | Liupanshui | Silver | Dazhou | Baoji | | | |
| Zhangye | An shun | Wuwei | Guang'an | | | | |
| Yulin | Silver | Zhangye | Jinchang | | | | |
| | Wuwei | | Silver | | | | |
| | Zhangye | | Wuwei | | | | |
| | Qingyang | | Yulin | | | | |
| □ | Yulin | □ | □ | □ | □ | □ | □ |

Spatial autocorrelation includes global spatial autocorrelation and local spatial autocorrelation analysis, which was conducted to assess the correlation between the common geographical features or attribute values in a regional unit and its neighbors. In previous studies, the focus has been on the spacetime evolution. Additionally, the spatial distribution factors and coupling coordination mechanism are rarely discussed [54]. In this paper, using Geo DA analysis, we tried to explore the different spatial-temporal characterized and distributed factors of land use benefits in western mining cities. The study showed that the land use benefits from spatial-temporal correlation characteristics, which were mainly affected by the node cities and the horizontal adjacency relation.

### 3.3.2. Local Spatial Autocorrelation Analysis of the Land Use Efficiency Coupling Coordination Degree

To assess the local agglomeration pattern of evolutionary characteristics and rules, a local indication of spatial association (LISA) agglomeration diagram of the land use efficiency of mining cities of western China was drawn by GeoDA analysis software [55] (Figure 6).

An analysis of the results indicates that during the study period, Shizuishan city, Erdos city, Jinchang city, and Wuhai city were the only cities that noticeably aggregated. Furthermore, the aggregation type of a city always changes over time. In 2000, Shizuishan city belonged to the significant type of "low-high", while Erdos city, Jinchang city, and Wuhai city belonged to the non-significant type.

In 2005, Shizuishan city changed to the non-significant type, Erdos was a "high-high" significant type, Jinchang city was a "high-low" significant type, and Wuhai city was still non-significant. In 2010, Erdos city was still a "high-high" significant type, while the other three cities were non-significant. In 2015, Erdos city and Jinchang city remained unchanged, while Shizuishan city and Wuhai city changed to the significant type of "high-high".

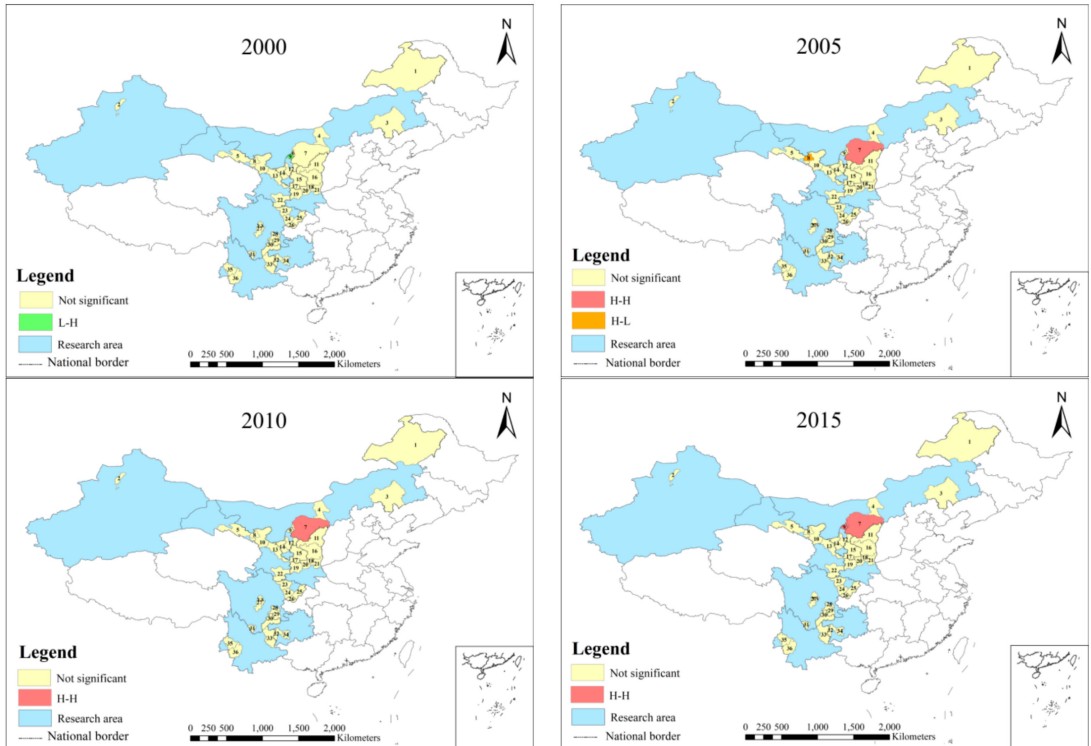

**Figure 6.** LISA cluster diagram of the coupling coordination degree for western China's mining cities in 2000, 2005, 2010, and 2015. In 2000 (bright green), 9: Shizuishan City; in 2005 (red areas), 7: Ordos City; (brownish yellow) 8: Jinchang City; in 2010 (red areas), 7: Ordos City; in 2015 (red areas), 6: Wuhai City, 9: Shizuishan City, and 7: Ordos City.

The characteristics of local aggregated evolution will now be discussed. The coupling coordination level of the land use benefit of the mining cities of western China differs significantly, especially for Baoshan city, Lincang city, Qujing city, Zhaotong city, Wuwei city, Wuzhong city, Zhongwei city, and Silver city, which differ from many other cities and have a low coupling coordination level. These cities showed the characteristic of "high-low" aggregation, with the neighboring city of Nanchong city. In 2000, the coupling coordination level of land use efficiency of Shizuishan city was higher than that of Erdos city, which was clearly affected by Shizuishan city, and both cites showed characteristics of "low-high" aggregation. Along with the promotion of the coupling coordination level of land use benefit of Erdos city, this area showed the characteristic of "high-high" aggregation in 2015. Moreover, Wuhai city had become a hotspot city with the promotion of the coupling coordination, and Jinchang city evolved from "high-low" aggregation to a non-significant type.

## 4. Conclusions

Based on the above analysis, the following main conclusions, limitations and future studies can be drawn:

(1)    Land use efficiency in the mining cities of western China gradually increased with relatively little variability and overall stability; land use efficiency had a staged distribution and showed different change trends with various time nodes. In 2010, the trends fluctuated noticeably;

(2) Spatial and temporal distribution characteristics of the coupling coordination degree: From 2000 to 2015, the land use efficiency coupling coordination degree ranged from 0.1620 to 5.503. There were three coupling coordination stages (reluctant, low, and moderate level). These were mainly dominated by reluctant and low levels of the coupling coordination degree. The coupling coordination level was relatively low. The different cities of coupling coordination developed produced temporal and spatial differences in the process of their evolution; Baotou city, Wuhai city, Erdos city, and Karamay city had a relatively high coupling coordination degree and played the leading role as the core cities;

(3) Spatial autocorrelation analysis of the coupling coordination degree: Land use benefits in mining cities of western Chia showed positive and negative spatial autocorrelations. In 2015, the spatial positive correlation among research units was relatively large. Shizuishan city, Ordos city, and Wuhai city had higher coupling coordination degrees, which is consistent with the result of spatial distribution characteristics (Figure 4). During the study period, only Shizuishan city, Ordos city, Wuhai city, and Jinchang city showed significant aggregation types. In 2005, 2010, and 2015, Ordos City was among the "high-high" aggregation type. Other cities had a low coupling coordination degree;

(4) In this paper, we selected a specific time of point as the study time scale, due to the limitation of data acquisition. The study on the long-term spatio-temporal evolution and law of land use benefits was affected. In future studies, the accuracy of indicators needs to be improved. Furthermore, MODIS data and Landsat data should be adopted, which can be explored from a land use classification perspective, and the long-term scale of the evolutionary mechanisms and reasons will be explored.

**Author Contributions:** Z.B. proposed the project support for this study; J.Y. designed the model, collected the data, and wrote the paper; Q.Y. and Y.P. polished the article. All of the authors have read and approved the final manuscript.

**Funding:** This research was funded by the National Special Project for Basic Science and Technology, "Investigation Project of Land Degeneration in Key Western Mining Areas" (Grant No. 2014FY110800).

**Acknowledgments:** The authors gratefully acknowledge the editors and anonymous referees for their comments regarding this study. We are grateful for support from the National Special Project for Basic Science and Technology.

**Conflicts of Interest:** The authors declare no conflict of interest.

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
