# Peer review of "Spatio-Temporal Distributions of the Land Use Efficiency Coupling Coordination Degree in Mining Cities of Western China"

_sustainability, doi:10.3390/su11195288_

Round 1
Reviewer 1 Report
The authors use the terms sub-systems and systems (Line 166: "the levels of the three land use efficiency sub-systems." ; Lines 178-179: "the three systems (society, economy, and environment)"; Lines 182-183 "the elements of three subsystems (economy, society, and ecology)"). It is necessary to clarify.
Line 189 : Formula (7) should be clarified (the variation for ∏ is not specified). See for example reference [30].
The author should clarify the notations in the following formulas: Lines 159-160: Formulas 5 and 6: the symbol wj or Wj (small letter or capital letter); Lines 213-217: Formulas 10 and 12: capital letter Wij.
Lines 213-217: Formulas 10 and 12: The symbol D bar should be explained.
The author should present in an explicit manner the coordinates of the two axes in the Moran scatter diagram (Lines 313-314).
Author Response
Response to Reviewer 1 Comments
Firstly, we appreciate you for your comments and suggestions, we all think the Comments are professional and they are really useful for us to receive and improve the article. According to your comments, we revised the article. The revised explanations are as follows:
Point 1 The authors use the terms sub-systems and systems (Line 166: "the levels of the three land use efficiency sub-systems."; Lines 178-179: "the three systems (society, economy, and environment)"; Lines 182-183 "the elements of three subsystems (economy, society, and ecology)"). It is necessary to clarify.
Response 1:Sub-systems include society benefits,economy benefits and environment benefits(Line158; Line170; Lines174-175); Evaluation system includes Criteria layer and Index layer, we revised them in the article (Line159).
Point 2 Line 189: Formula (7) should be clarified (the variation for ∏ is not specified).
Response 2: It is easier to understand to change the formula and add references (Lines181), I hope that you’re satisfied with the revisions.
Point 3The author should clarify the notations in the following formulas: Lines 159-160: Formulas 5 and 6: the symbol wj or Wj (small letter or capital letter); Lines 213-217: Formulas 10 and 12: capital letter Wij.
Response 3: Formulas 5 and 6: the symbol wj is small letter (Lines151-156); Formulas 10 and 12: Wij is capital letter(Line204-208), they express different meanings. Formulas 5 and 6, wj represents the index weight. Formulas 10 and 12, Wij represents the spatial weight coefficient matrix(Lines208), I hope the explanation can solve your doubts.
Point 4 Lines 213-217: Formulas 10 and 12: The symbol D bar should be explained.
Response 4:`D represent the mean value of the coupling coordination degree of space units i and j.(Lines207)
Point 5 The author should present in an explicit manner the coordinates of the two axes in the Moran scatter diagram (Lines 313-314).
Response 5: Thank you for your suggestion, we redrew the scatter plot, and explained the two axes(Lines 329-331). (Horizontal axis) LUB_CCD represents the land use benefit of Coupling coordinated degree; (Vertical axis) W_ LUB_CCD is spatial lag factor for land use benefit of Coupling coordinated degree (Lines332-334), I hope the explanation can solve your doubts.

Reviewer 2 Report
Dear all,
the manuscript presents an interesting crossing of the Spatio-temporal distributions of land use and the issue of mining cities. The manuscript presents interesting scientific soundness as well as a simple but efficient methodology.
However, some improvements should be made before publication. The discussion and conclusions should present similar studies and researches (referenced) as well as a section regarding study limitations and further research lines should be added.
best,
Author Response
Response to Reviewer 2 Comments
Firstly, we appreciate you for your comments and suggestions, we all think the Comments are professional and they are really useful for us to receive and improve the article. According to your comments, we revised the article. The revised explanations are as follows:
Point However, some improvements should be made before publication. The discussion and conclusions should present similar studies and researches (referenced) as well as a section regarding study limitations and further research lines should be added.
Response: we appreciate you for your comments and suggestions, Thank you for your approval of the article's methods and research ideas. According to your suggestion, we have added and modified the discussion, conclusion part and the references, the revised explanations are as follows:
(1)Added and modified the discussion (Lines240-249; Lines306-313; Lines358-365)
In previous studies, The traditional entropy method often leads to a deviation of the evaluation results due to extreme or negative values in the evaluation [51] .In this study, Compared with traditional methods, the improved entropy method is applied to measure the land use benefit Coupling Coordination Degree in Mining Cities of Western China, Which compensate for the defect of the traditional entropy method through standardization processing. This method can objectively and effectively analyze the information relevance between index variables and avoid the limitation of subjective weighting to a great extent. In addition, traditional method is usually represented by a radar chart [52, 53].In this study, Origin software was used to draw the accumulation map of land use efficiency. It is easy to present an overview of evolution character, and analyze the relationship of society benefits, economy benefits and environment benefits(Lines240-249).
The previous studies mainly present the object of the spatio-temporal distribution [54,55].In this paper, with the ArcGIS spatial analysis software and Coupling coordination model, Which explores the spatial-temporal differentiation characteristics of coupling coordination level of land use benefits in western mining cities. The paper also studies the degree of economic, social and ecological coupling coordination of the factors influencing the coupling coordination level of land use benefit in each research unit. Then the importance of the land use ecological and environmental benefits, and the significance of carrying out the land use comprehensive, sustainable and harmonious development of economy, society and ecology(Lines306-313).
Spatial autocorrelation includes Global spatial autocorrelation and Local spatial autocorrelation analysis, which was conducted to assess the correlation between the common geographical features or attribute values in a regional unit, and its neighbors. In previous studies, just focus on the Space-time evolution. While, the spatial distribution factors and coupling coordination mechanism are rarely discussed [57-61]. In this paper, with the Geo DA analysis was try to explore the spatial-temporal different characterized and distributed factors of land use benefits in western mining cities. The study showed that the land using benefit from spatial-temporal correlation characteristics. It mainly affected by the node cities and the horizontal adjacency relation (Lines358-365).
(2) Added and modified the references (Lines475-480; Lines 526-528; Lines556-581)
(3)Added and modified the conclusion (Lines420-425)
In this paper, we selected a specific time of point as the study time scale, due to the limitation of data acquisition. The study on the long-term spatio-temporal evolution and law of land use benefits were affected. In future studies, the accuracy of indicators are needs to be improved. Furthermore, the MODIS data and Landsat data should be adopted, which can be explored from land use classification perspective, and the long time scale of the evolution mechanisms and reasons will be explored(Lines417-422).

This manuscript is a resubmission of an earlier submission. The following is a list of the peer review reports and author responses from that submission.
Round 1
Reviewer 1 Report
This manuscript focuses on the spatio-temporal pattern of land use efficiency coupling coordination degree in mining cities in Western China. The research topic is of great importance. However, this article has the following problems that need to be addressed.
(1) The manuscript is not well written. It is difficult to read through. Some sentences are awkward and wordy. There are also grammatical errors and omissions throughout the article. If the author(s) are non-native English speakers, they might need English proofreading and language editing before the submission. Please make sure the manuscript meets the requirements of academic writing: clear, precise, and consistent.
(2) The Introduction section does not provide enough background. As the most important concept for this article, coupling coordination degree is not clearly defined in the first place. The author(s) seem to be quite familiar with the related literature. But they fail to succinctly and effectively provide a literature review. Also, technical terms are used without a brief explanation. For example, “Research methods mainly include the entropy method, AHP, the coefficient of variation method, GM (1, 1) and principle component analysis” (Line 51). The abbreviations AHP and GM (1,1) are used as they are without any explanation.
(3) Data and several figures are not well represented. The Data and Methods section includes data sources and data standardization but fails to specify what exactly the data are. They are later lumped together in a hard-to-recognize manner in Table 2 in the Results and Discussion section. Figure 1 and Figures 4-7 are too small to recognize. The audience can hardly read the map legend let alone the spatio-temporal distributions. The authors might consider using points on the map and numbers in the notes to represent the location and name of mining cities rather than their full names, which fail to accurately represent their location and make the figures look overcrowded.
(4) Analyses of the spatial and temporal characteristics of land use efficiency are too simplistic and overgeneralized. “First, the overall land use efficiency increased over the study period with low variability within each city (Figure 3)”(Line 142). However, based on Figure 3 there are several different temporal patterns of land use efficiency that can be identified in these mining cities, not “increased over the study period with low variability within each city”. “Coupling coordination degree was divided into three types: reluctant, low and moderate” (Line 152-156). Baotou and Karamay achieved the moderate coupling coordination degree in 2010. The authors fail to mention this in “ â‘¢ Moderate coupling coordination (0.5-0.8)” (Line 156). It is hard to see the spatial distribution of coupling coordination degree because the small size of Figures 4-7. The analyses on Pages 10-12 are simplistic description on how coupling coordination degree in major cities of the three regions changed over the 2000-2015 study period, not exactly in the spatial dimension.
(5) Last sentence in the Abstract “There was an obvious ‘center-periphery’ (core-periphery, reviewer’s correction) gradual step-down (trickle-down, reviewer’s correction) trend, matching the four phases of development in the resource-based cities” (Line 22-23) has never been discussed in the text. Also, no cities in this study have gone through the four stages identified in Table 4 (Page 14).